# Efflux pump-mediated resistance to new beta lactam antibiotics in multidrug-resistant gram-negative bacteria
Augusto Dulanto Chiang[1,3] & John P. Dekker [1,2] ✉

The emergence and spread of bacteria resistant to commonly used antibiotics poses a critical threat to modern medical practice. Multiple classes of bacterial efflux pump systems play various roles in antibiotic resistance, and members of the resistance-nodulation-division (RND) transporter superfamily are among the most important determinants of efflux-mediated resistance in gram-negative bacteria. RND pumps demonstrate broad substrate specificities, facilitating extrusion of multiple chemical classes of antibiotics from the bacterial cell. Several newer beta-lactams and beta-lactam/beta-lactamase inhibitor combinations (BL/BLI) have been developed to treat infections caused by multidrug resistant bacteria. Here we review recent studies that suggest RND efflux pumps in clinically relevant gram-negative bacteria may play critical but underappreciated roles in the development of resistance to beta-lactams and novel BL/BLI combinations. Improved understanding of the genetic and structural basis of RND efflux pump-mediated resistance may identify new antibiotic targets as well as strategies to minimize the emergence of resistance.

The development of antibiotics is a key achievement of modern medicine. In addition to their direct therapeutic use in the management of serious bacterial infections, antibiotics have made advances in chemotherapy, transplantation, and surgery possible, which depend on the ability to control secondary bacterial infections that can occur when immune defenses are compromised. The clinical use of antibiotics, however, can generate selection pressures on the underlying treated bacterial populations that drive the evolution of resistance through a variety of mechanisms[1]. The emergence of antimicrobial resistance (AMR) within bacterial populations has resulted in the broad loss of efficacy of many commonly used antibiotics, and the spread of highly drug resistant bacteria poses a critical threat to public health.

Gram-negative bacteria such as *Pseudomonas aeruginosa*, *Acinetobacter baumannii-calcoaceticus*, *Klebsiella pneumoniae*, and other members of the order Enterobacterales (see Box 1 for a glossary explaining the technical terms used) can cause serious infections that result in significant morbidity and mortality. One of the most successful classes of antibiotics used to treat infections caused by gram-negative bacteria are the beta lactams, which act by inhibiting the DD-transpeptidase enzymes that cross-link the cell wall that is essential for viability. Thus, resistance to beta lactam antibiotics in gram-negative bacteria is of critical importance. Beta-lactam resistance in gram-negative bacteria, in turn, is generally driven by a combination of four major mechanisms: drug hydrolysis by beta-lactamases; alteration of target sites in the penicillin-binding proteins (DD-transpeptidases); alteration of drug permeability, primarily through the modification of outer membrane porin channels; and extrusion from the cell through the activity of efflux pumps[2,3] (Box 2).

Over the past decade, several new beta-lactams and beta-lactam/beta-lactamase inhibitor combinations (hereafter described as novel BL/BLI) have been developed to meet the increasing threat of carbapenemase-producing organisms (CPO)[4–6]. Carbapenemases are beta lactamases that exhibit activity against carbapenems and most other beta lactams, rendering the bacteria that produce these enzymes resistant to these agents. The addition of broad spectrum carbapenemase inhibitors in novel BL/BLI combinations can restore susceptibility to the included beta-lactam, and thus these combination antibiotics are often used as a treatment for CPO infections[7]. These agents have also shown efficacy against other difficult to treat pathogens, such as *P. aeruginosa*, *Stenotrophomonas maltophilia* and carbapenem-resistant *A. baumannii-calcoaceticus* complex (CRAB) that become resistant to carbapenems through mechanisms other than acquired carbapenemases[8,9]. Unsurprisingly, in the years following the introduction of these new antibiotics into clinical practice, resistant bacteria have been increasingly reported. Multiple studies have described genomic changes associated with this resistance[10–21], and a smaller subset have provided experimental validation of the effects of specific mutations[22–27] (Box 3). Many of the observed genetic changes involve RND efflux pumps, some of which are currently under-characterized, but likely play important roles in

[1]Laboratory of Clinical Immunology and Microbiology, NIAID, NIH, Bethesda, MD, USA. [2]National Institutes of Health Clinical Center, NIH, Bethesda, MD, USA. [3]Present address: Vanderbilt University Medical Center, 1211 Medical Center Drive, Nashville, TN, 37232, USA. ✉e-mail: john.dekker@nih.gov

## Box 1 | Glossary

**Aminoglycoside:** Class of natural and semi-synthetic antibiotics based on amino-modified sugars that are used in the treatment of infections caused by gram-negative bacteria and some gram-positive bacteria. Examples include gentamycin, amikacin, and tobramycin.

**AmpC cephalosporinase:** A family of beta lactamase enzymes that can hydrolyze cephalosporin antibiotics.

**Beta lactam:** Class of natural and semi-synthetic antibiotics that contain a four membered cyclized amide ring. Beta lactam antibiotics work by inhibiting the DD transpeptidases that cross-link the bacterial cell wall. Classes of beta lactams include penicillins, aminopenicillins, cephalosporins, monobactams, and carbapenems.

**Carbapenem:** Class of broad-spectrum beta lactam antibiotics often used to treat infections caused by gram-negative and gram-positive organisms that are resistant to other beta lactams. Examples include ertapenem, imipenem, and meropenem.

**Cephalosporin:** Class of broad-spectrum beta lactam antibiotics that are used to treat infections caused by gram-negative and gram-positive bacteria. Examples include ceftazidime, ceftriaxone, and cefepime.

**Enterobacterales:** A medically important order of gram-negative facultatively anaerobic bacilli. Members include *E. coli*, *Klebsiella pneumoniae*, and *Enterobacter* sp.

**Fluoroquinolone:** Class of synthetic antibiotics that work by inhibiting topoisomerase enzymes that are used to treat gram-negative and gram-positive infections. Examples include ciprofloxacin, levofloxacin, and moxifloxacin.

**Frame shifting indel:** An insertion or deletion in the genetic sequence coding for a protein that alters the translation reading frame, resulting in a protein product with an incorrect amino acid sequence.

**Genomic variants:** Alterations in the genome of an organism that may include simple mutations that change the base at a position, and more complex alterations that include deletions, insertions, inversions, and translocations.

**Monobactam:** Class of beta-lactam antibiotics used to treat infections caused by gram-negative organisms. Aztreonam is the member of this class currently used in clinical practice, and it is often used in patients with severe allergies to other beta-lactam antibiotics.

**Sulfonamide:** Class of synthetic antibiotics that act by interfering with bacterial folate metabolism. The sulfonamide antibiotic sulfamethoxazole is used in conjunction with trimethoprim to treat a variety of gram-negative and gram-positive bacterial infections.

**Tetracycline:** Class of natural and semi-synthetic antibiotics used to treat infections caused by both gram-positive and gram-negative bacteria. Examples include tetracycline, doxycycline, and minocycline.

## Box 2 | Mechanisms of beta lactam resistance

Beta lactams are enzyme inhibitors produced by bacteria and fungi. These molecules, and various human-made semi-synthetic beta lactam antibiotics that are derived from them, inhibit the DD transpeptidases (also called penicillin binding proteins, or PBPs) that cross-link the bacterial cell wall. Loss of cross-linking results in cell lysis and death in response to osmotic and other physical stresses. Because of the abundance of beta lactams, diverse mechanisms have evolved to reduce the impact of these compounds. The outer cell membrane is relatively impermeable to beta lactams, and they may enter the cell through protein channels called outer membrane porins (OMPs). OMP deletions or other mutations that modify the permeation pathway and reduce beta lactam entry are a first line of defense and significant contributor to resistance. Beta lactams that make it into the periplasmic space or cytoplasm may be extruded from the cell by efflux pumps or hydrolyzed and inactivated by a family of enzymes called beta lactamases. Finally, the beta lactam binding site of the DD transpeptidase itself can overexpressed or mutated in a variety of ways to decrease affinity for specific beta lactams and confer resistance.

## Box 3 | Experimental verification of resistance mutations

Mutations associated with antibiotic resistance often occur in complex backgrounds and occur alongside many other genomic changes. It is thus usually only possible to demonstrate that specific individual mutations cause resistance experimentally. A well-used experimental approach involves engineering mutations individually into a bacterial strain with a defined genetic background to create an isogenic pair consisting of the parent strain and a derivative strain that differs only in the mutation being investigated. The minimum inhibitory concentration (MIC) of the antibiotic in question is then tested for both isolates in the pair and the difference in MIC can be attributed to the single mutation that differs between the two isolates. Groups of mutations can also be tested in different combinations to establish synergistic relationships.

Bl/BLI resistance. Unfortunately, there are currently no standard approaches for specific molecular or functional testing for efflux-mediated resistance in clinical microbiology laboratories, and while efflux pump inhibitors are under development, none are currently available for clinical use.

In this Perspective, we briefly introduce the structure and role of RND efflux pumps in bacteria. We then review recent work that highlights the importance of RND efflux pumps in resistance to novel BL/BLI in clinically relevant gram-negative bacteria. The novel BL/BLI combinations currently used, or in clinical development that will be covered, as well as the year of FDA approval, are ceftazidime/avibactam (CZA, 2015), ceftolozane/

tazobactam (C/T, 2014), meropenem/vaborbactam (MEM/VAB, 2017), imipenem/relebactam (IMI/REL, 2019), cefepime/zidebactam (FEP/ZID, not approved, but available under compassionate use since 2021) and sulbactam/durlobactam (SUL/DUR, 2023). We will additionally cover cefiderocol (FDR, 2019), a new siderophore cephalosporin and last-line agent in clinical use.

### RND efflux pumps and their regulators are commonly mutated in Gram-negative bacteria with AMR

RND efflux pumps play essential roles in bacterial physiology, facilitating the extrusion of a wide range of metabolic and organic products, including

molecules involved in quorum sensing[28,29], the production of biofilms[30], and pathogenesis and virulence[31,32]. These pumps are organized as large tripartite complexes that span the gram-negative envelope from inner to outer membrane, usually consisting of homomeric assemblies of three different components: an inner membrane pump (IMP), membrane fusion protein (MFP) and an outer membrane porin (OMP)[33] (Fig. 1a). While most characterized RND pumps can transport a wide variety of chemically diverse substrates, individual pumps have very different substrate preferences, which have clinical consequences for antibiotic resistance[34–38]. For example, in *P. aeruginosa*, the MexAB-OprM efflux pump is able to transport beta-lactams, fluoroquinolones, and sulfonamides, in addition to a highly diverse set of metabolites, while the MexXY-OprM pump preferentially transports aminoglycosides and tetracyclines[39]. Mutations in the efflux pump regulatory systems leading to pump overexpression are among the most common efflux-related mechanisms for drug resistance described[40]. Meanwhile, amino acid substitutions in known structural binding sites in the IMP subunits of RND efflux pumps appear to have the greatest impact on substrate specificity[36,37,41].

## BL/BLI resistance in *P. aeruginosa* is mediated by multiple chromosomal RND Efflux Pumps

*P. aeruginosa* is one of the most important human pathogens causing infections in hospital settings and in immunocompromised hosts[42]. Unique characteristics of this organism include its highly impermeable envelope relative to Enterobacterales[43,44], and the presence of a large number of chromosomal resistance genes, including at least 12 distinct RND-type efflux systems[45]. This allows it to acquire resistance to all classes of currently used antibacterial agents through chromosomal mutations. The RND efflux pumps that are mutated display broad substrate specificities demonstrating significant overlap, as has been reviewed in detail elsewhere[36,46–48]. The major efflux pump, MexAB-OprM, plays a critical role in the development of clinical resistance against penicillins, cephalosporins, monobactams, meropenem, fluoroquinolones, macrolides, and sulfonamides, and the consequences of amino acid substitutions in MexB have been the subject of extensive study[37,48–50]. Additional notable pumps known to contribute to clinical resistance include MexCD-OprJ, involved in resistance to cefepime, macrolides, and fluoroquinolones[46,51]; and MexXY-OprM, which contributes to aminoglycoside, tetracycline, macrolide, and fluoroquinolone resistance[46]. Other less well-described pumps in the context of antibiotic resistance include MexEF-OprN, MexGHI-OpmD, MexMN-OprM, MexPQ-OpmE, MexVW-OprM, MuxABC-OpmB and TriABC-OpmH.

The novel BL and BL/BLI alternatives currently in use or development to treat carbapenem-resistant *P. aeruginosa* include CZA, C/T, IMI/REL, FEP/ZID, and FDR. The most common mechanisms of CZA and C/T

resistance in *P. aeruginosa* are mutations in the chromosomal AmpC cephalosporinase (PDC), often located in the omega loop region of the enzyme[52,53], or increased expression of MexAB-OprM[12,54,55]. Other efflux pumps are likely to be contributing as well. For example, in an in vitro experimental evolution system, it was found that in increased expression of the MexVW efflux pump due to an intergenic mutation upstream of *mexV*, in addition to a MexW E36K substitution, can contribute to resistance both to CZA and C/T, and to cefepime (FEP)[54]. These mutations spontaneously emerged in a DNA mismatch repair-deficient hypermutator background with a non-functional MexAB-OprM, and their role in conferring a 4- to 6-fold increase in minimum inhibitory concentration (MIC) to CZA, C/T and FEP was confirmed through genetic engineering in a wild type background. Of note, MexVW is more phylogenetically distant than the other major *P. aeruginosa* RND efflux pumps[37,41], and while its ability to confer resistance to fluoroquinolones, tetracycline, chloramphenicol, erythromycin, and cefpirome in vitro has been previously demonstrated[56], little attention has been paid to this pump in the context of clinical antibiotic resistance, particularly to novel BL and BL/BLI during the past decade[57].

A variety of genomic variants were observed to emerge in *P. aeruginosa* laboratory strains in a study involving serial passaging with ceftazidime (CAZ) or CZA[23]. In addition to mutations occurring in relation to MexAB-OprM that were common to both antibiotic exposures, variants observed only in the CZA group included mutations associated with the MexMN-OprM efflux pump and its upstream regulators that emerged in independent lineages (Table 1). MexMN-OprM may play a role in susceptibility to diverse classes of antibiotics[58], and increased expression of MexMN can contribute to resistance to imipenem and to the D13-9001 efflux pump inhibitor[59]. However, establishing the precise role of MexMN-OprM in resistance to new antibiotics requires further study including testing the effects of individual mutations in appropriate genetic backgrounds.

The potent antipseudomonal activity of C/T is thought to result largely from its ability to accumulate in the periplasmic space because ceftolozane is a poor substrate for MexAB-OprM. Thus, while multidrug-resistant *P. aeruginosa* that have been studied commonly overexpress this major RND efflux pump[12,60], C/T is minimally affected by this mechanism. Mutational C/T resistance in *P. aeruginosa* has largely been described to emerge from substitutions in the PDC beta-lactamase[61]. Nonetheless, evidence is emerging demonstrating that mutations and/or overexpression of other efflux pumps can contribute to C/T resistance. For instance, a clinical isolate was described with disruptions in NfxB, a repressor of MexCD-OprJ; MexD S133G, which forms part of the serine loop; and MexD Q178R, predicted to be immediately adjacent to a short loop lining the pump's distal binding pocket[25]. The role of MexCD-OprJ in C/T and CZA resistance was confirmed experimentally[25]. As noted above, substitutions resulting in increased

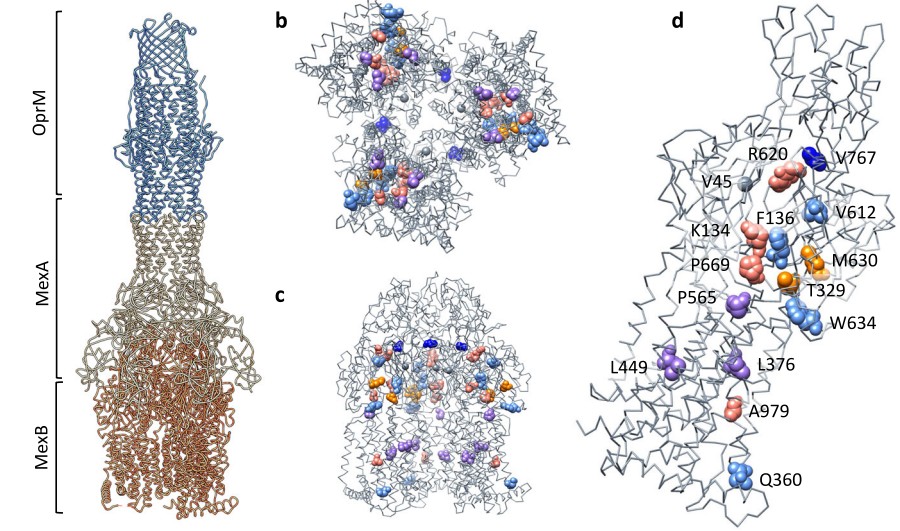

**Fig. 1 | Structure of a canonical RND efflux pump tripartite complex and residues important to antibiotic resistance. a** Structure of the complete MexAB-OprM complex from *P. aeruginosa* (rendered from PDB Accession 6IOL using UCSF Chimera). OprM is the outer membrane porin in the complex and is a homotrimer. MexA is the membrane fusion protein and is a homohexamer. MexB is the inner membrane pump and is a homotrimer. **b** MexB trimer viewed from the intracellular face with residues associated with antibiotic resistance from Table 1 shown in spacefill representation. Colors are defined at end. **c** MexB trimer viewed parallel to inner membrane. **d** MexB monomer with individual residues corresponding to Table 1 labeled. Colors indicate residues associated mutations conferring resistance to the following agents: Purple = CZA; Dark blue = both CZA and FDR; Pink = IMI/REL; Orange = both IMI/REL and FEP/ZID; Light blue/gray = FEP/ZID.

**Table 1 | Substitutions in RND Efflux pumps associated with resistance to novel BL/BLI in _P. aeruginosa_ reported in the literature**

| Efflux pump | CZA (2015) | C/T (2014) | IMI/REL (2019) | FEP/ZID (2021*) | FDR (2019) |
|---|---|---|---|---|---|
| MexAB-OprM | **MexA**[12,23], **MexB**[12] L376V, L449F, P565S, V767G[23] **MexR**[12,54] | | **MexB** K134N[19], T329A[24], R620C[19,24], G621S[19], M630V[24], P669L[24], A979T[24] **MexR** L57Q, R83C[19], S92G, L54P, L57P[24] | **MexB** V45L, F136L, T329A, R360C, V612M, M630T, W634G[26] | MexA[20] MexB V767G[21] MexR[17,21] |
| MexCD-OprJ | **MexD** S133G, Q178R[25] **NfxB** F/S[25] | **MexD** S133G, Q178R[25] **NfxB** F/S[25] | | | MexC[20] |
| MexEF-OprN | MexE[12] | | **MexF** V497A, F/S[19] **MexE** F/S[19] **MexT** S41G, F/S, G148R[19] **MexS** P254S[19] | MexF Y845H[26] | MexE, MexS[20,21] |
| MexGHI-OpmD | | | MexH E104K[24] | | |
| MexJK | | | MexK D616G[24] | | |
| MexMN-OprM | MexM[12] **MexN**[12]A44V[54], A277V, P279L, S334L[23] PA14_45870 F/S[23] PA14_45880 M53I[23] | | | | |
| MexVW | **MexW** E36K[54], overexpression | **MexW** E36K[54], overexpression | MexW V444A[24] | | |
| MexXY-OprM | MexXY (large deletion)[23] MexZ[23] | | MexY D237G, E268K[24] MexZ F/S[14] | MexY[26] | MexY, MexZ[20] |

_CZA_ ceftazidime/avibactam, _C/T_ ceftolozane/tazobactam, _IMI/REL_ imipenem/relebactam, _FEP/ZID_ cefepime/zidebactam, _FDR_ cefiderocol, _F/S_ frameshifting mutation.
Protein names and/or substitutions in bold face indicate relatively strong evidence, defined as clonal isolates with MIC disparity, serial passaging with recurring mutations over different lineages, or mutation engineering experiments. Dates in parentheses indicate FDA approval. *FEP/ZID is not currently FDA approved at the time of this writing but has been clinically available under compassionate use since 2021.

expression and modifications of MexVW have also been demonstrated experimentally to result in increases in C/T MIC[54].

IMI/REL is another potent agent for the treatment of carbapenem-resistant _P. aeruginosa_. A major mechanism of resistance to carbapenems in _P. aeruginosa_ is disruption of the major outer membrane porin OprD[44]. Increased efflux through the MexAB-OprM efflux pump, particularly when overexpressed, plays an additional role in MEM and MEM/VAB resistance, while IMI is less affected by MexAB-OprM efflux, as it is a worse substrate of this major efflux pump. Nonetheless, IMI can still be hydrolyzed to some extent by the _P. aeruginosa_ PDC[62]. This has been suggested to explain why the addition of REL to IMI restores susceptibility in about 80% of carbapenem-resistant _P. aeruginosa_[62], while the combination of VAB and MEM does not appear to perform significantly better than MEM alone in a collection of similar clinical isolates[63].

A recent study has provided insight into how IMI/REL resistance develops in _P. aeruginosa_. Serial whole genome sequencing of clinical _P. aeruginosa_ isolates that developed increased IMI/REL MIC from five patients with ventilator-associated pneumonia treated with IMI/REL for 10–28 days found most isolates harbored disrupted OprD porins at baseline[19]. A pattern of emerging mutations in MexB, MexR (a major MexAB-OprM repressor), and frameshifting mutations in MexEF-OprN components or its regulators (Table 1 and Fig. 1b-d) was then observed. While individual validation of these variants was not performed, the shared targets across these independent lineages, and the genomic data associating the genetic changes to MIC increases provides supportive evidence of the involvement of the MexAB-OprM efflux pump in IMI/REL resistance. Among the observed variants, three MexB mutations – K134N, located in the serine loop; and G621S and R620C, located in the switch loop – were situated in the lining of the pump's ligand binding pockets, further supporting a prediction of functional effect[19]. Most of the mutations in MexEF-OprN or its regulator MexS/T in this report (Table 1) are consistent with frameshifting indels, which would be predicted to result in truncated translation products[19]. The MexAB-OprM and MexEF-OprN pumps are co-regulated[47,64], and it has previously been shown that disruption of MexEF-OprN can result in increased expression of MexAB-OprM[47], suggesting that this increased expression may contribute to the mechanism conferring resistance. As the authors point out, given that IMI is not efficiently transported by MexAB-

OprM, it is possible that increased efflux of REL by MexAB-OprM plays a role[19]. The role of MexAB-OprM overexpression in IMI/REL resistance in clinical isolates has been suggested in another recent case report demonstrating divergent evolution of resistance to CZA, C/T, and IMI/REL in a set of clinical isolates from a patient treated with CZA[65].

However, at least one previous study found that overexpression of wild-type MexAB-OprM had little effect in IMI/REL MIC[66], suggesting the possibility that MexB substitutions that modify specificity may be required for MexAB-OprM to confer resistance. In a different study, several _P. aeruginosa_ laboratory strains, along with isogenic MutS-deficient hypermutator counterparts, underwent serial passage through increasing concentrations of IMI/REL for 7 days[24]. In addition to OprD inactivation, overexpression of MexB and mutations in MexR and MexB were observed in several lineages (Table 1)[24]. These mutations included MexB R620C, corroborating clinical variants found by Shields and colleagues and discussed above[19]. Out of the five MutS-deficient hypermutator lineages evolved, two were also found to develop MexY mutations, and one each had MexF, MexK, MexH and MexW variants[24]. As the functional consequences of these variants were not verified experimentally, establishing their significance requires further study.

FDR is a newer promising agent which demonstrates potent activity against carbapenem-resistant gram-negatives, including _P. aeruginosa_ and organisms that harbor metallo-β-lactamases. FDR is a siderophore antimicrobial which exploits the bacterial iron transport mechanisms to gain entry into the cell and is stable against hydrolysis by most beta-lactamases[67]. The first resistance mechanisms described were mutations in genes encoding iron transport system components, including _piuA, piuB, pvdS, pirS, pirR, tonB3, exbB,_ and _exbD_, consistent with its mode of transport[17,20,21,68,69]. Early in vitro studies suggested a minor role for MexAB-OprM, with a maximum of 2-fold increase in MIC in laboratory strains with MexAB overexpression[70]. Separately, multiple mutations outside iron transport systems that may be associated with FDR resistance were identified in clinical isolates from individuals with cystic fibrosis[20]. Multiple efflux genes were mutated in these isolates including _mexZ_ (a MexXY-OprM regulator), _mexY, mexA, mexC,_ and _mexS_ (a MexEF-OprN regulator). Four _mexA_ mutations in 4/6 resistant isolates were predicted to be disruptive or inactivating. The set of isolates above contained additional

mutations in other known resistance mechanisms including the PmrAB two-component system, the UTP-glucose-1-phosphate uridylyltransferase, PBP2, PBP3 (DD-transpeptidases), the major porin OprD, and PDC. The relative potential roles of these individual mutations were not validated experimentally. Other studies examined the genomic changes in serial clinical isolates that developed FDR resistance, and found mutations in *mexR*, *mexE* and *mexB* (MexB V767G) (Table 1)[17,21]. As above, the specific roles of these efflux system mutations in FDR resistance were not confirmed experimentally, and thus would require further study[17,20,21].

FEP/ZID is a novel BL/BLI with additional antimicrobial activity attributable to intrinsic PBP2 affinity of the BLI; FEP/ZID has thus been described as a beta-lactam/beta-lactam enhancer agent[71,72]. The combination is currently in late stages of development and has demonstrated in vitro activity against metallo-beta-lactamase producing gram-negative bacteria[73]. Genomic changes associated with resistance to FEP/ZID have been described in *P. aeruginosa* laboratory strains and isogenic MutS-deficient hypermutator counterparts[26]. Among others, mutations in the MexAB-OprM pump and its regulators emerged independently in different lineages, including several mutations in key residues lining the ligand binding pockets (F136L, T329A, V612M, M630T). In addition, increased expression of *mexB* was confirmed in all lineages with increased MIC, further supporting a role of this pump in FEP/ZID resistance. Less uniformly, some lineages developed overexpression of *mexY* and mutations in *mexF* (Table 1).

## Overexpression and alterations of AcrAB-TolC may play roles in BL/BLI resistance in Enterobacterales

The prototype RND efflux pump in Enterobacterales, AcrAB-TolC, plays a major role in resistance to almost every antibiotic class including the beta-lactams, with the notable exception of aminoglycosides[47,74]. AcrD, as part of AcrAD-TolC, plays an important role in aminoglycoside efflux[75]. In many Enterobacterales, AcrAB-TolC is under the control of several regulators including the local repressor AcrR and global transcriptional factors MarR, MarA, RamR, RamA, SoxS, and Rob[76–78]. Some of these factors counter-regulate both AcrAB-TolC and outer membrane porins important for beta-lactam transport, including OmpC and OmpF in *E. coli*, and OmpK36 and OmpK35 in *K. pneumoniae*[79,80]. A combination of efflux pump expression and porin loss often confers carbapenem resistance in clinical Enterobacterales isolates that lack carbapenemases[81]. Several additional RND efflux pumps are present in Enterobacterales, including AcrEF and MdtABC in *E. coli*, and OqxAB and KpgABC in *K. pneumoniae*, but the roles of these additional pumps in antibiotic resistance, as well as those of others outside the RND family, have not been studied in great depth[82,83].

CZA resistance in carbapenemase-producing Enterobacterales has been associated with substitutions in class A carbapenemases, particularly in the KPC enzyme[84]. Additionally, some reports on KPC-containing *K. pneumoniae* suggest a possible role of AcrAB-TolC overexpression and/or porin deficiencies in CZA resistance[11,85], but other studies suggest the contribution of efflux may be minor[86]. AcrAB-TolC overexpression and porin defects were frequent in a collection of novel BL/BLI-resistant Enterobacterales that lacked carbapenemases[15], but the contributions of these changes to resistance were not directly examined experimentally in this study. Though a number of studies have investigated expression levels of the AcrAB-TolC efflux pump, less is known about the roles of specific coding variants in resistance to BL and BL/BLI[16]. In one study, genomic sequencing of two serial clinical KPC-containing *Enterobacter cloacae* clinical isolates that developed CZA resistance during treatment found an AcrB F396L substitution as one of 6 unique variants differentiating the susceptible and resistant isolates[18]. These changes in aggregate were associated with a 32-fold increase in the CZA MIC (from 1 to 32 μg/mL). The F396 residue is located in the TM4 domain of AcrB, and is 70-80% conserved across AcrB homologs[37]. In a separate in vitro study, Enterobacterales isolates containing extended spectrum beta lactamases including CTX-M were exposed to increasing CZA to evolve resistance, followed by genomic sequencing[22]. Among 19 isolates with increased CZA MIC, the only change observed in two CTX-M-15-expressing *E. coli* isolates was an AcrB F615S substitution,

conferring an 8- to 16-fold increase in CZA MIC. The F615 residue is notable for being part of the well-characterized phenylalanine-rich site lining the distal binding pocket involved in direct substrate interactions[37,87] suggesting that this substitution might alter substrate affinity, binding kinetics, or specificity. However, the effect of these isolated substitutions was not experimentally validated.

A second agent clinically used in carbapenem resistant Enterobacterales (CRE) infections is MEM/VAB. Increased AcrAB-TolC expression has been found among KPC-producing Enterobacterales with elevated MEM/VAB MIC and no metallo-β-lactamases[10] as well as among carbapenemase-negative CRE[15]; however, the role of efflux in MEM/VAB resistance is not clear. In a different study, the change in MEM/VAB susceptibility in *E. coli* and *K. pneumoniae* strains was tested with various engineered resistance mechanisms, including carbapenemases of the four Ambler classes, porin disruptions and RamR inactivation with consequent AcrAB-TolC over-expression and OmpK35 downregulation[88]. In this study, the effect of RamR inactivation was comparable to OmpK35 inactivation alone and was minimal in the background of OmpK35 and/or OmpK36 inactivation. This implies that, while porins are likely to be involved in VAB entry to the cell, the effect of AcrAB-TolC upregulation did not seem to contribute meaningfully to MEM/VAB resistance.

FDR is another agent that demonstrates broad in vitro activity against most CRE[70,89–91]. Currently it is not usually the first choice of therapy as there is less clinical experience with it, and other options are available for most serine carbapenemase-producing Enterobacterales, such as CZA and MEM/VAB[67,92]. FDR might be most useful in the context of metallo-β-lactamase producing Enterobacterales for which other good options are scarce[93]. The FDR resistance mechanisms are still being characterized in Enterobacterales. Several in vitro and in vivo studies point to TonB-dependent iron transport mechanisms, including CirA and Fiu[94,95], while others have reported substitutions/deletions in the AmpC beta-lactamase R2 loop[13,96] or the contribution of KPC alleles that confer CZA resistance[97]. An examination of the genetic makeup of fourteen clinical isolates that developed increases in FDR MIC during the APEKS-NP and CREDIBLE-CR clinical trials[98–100] identified changes in AmpC as a candidate resistance mechanism in one *Enterobacter cloacae* isolate, while no other changes in iron transport genes, outer membrane porins or beta-lactamases were identified in four other Enterobacterales isolates. Of note, in this study the increase in MIC remained unexplained in 10/14 isolates, and efflux pump expression or coding sequence variants were not explicitly analyzed. Overall, data on the impact of efflux pump variants or overexpression in FDR resistance in Enterobacterales remain scarce.

## The role of efflux-mediated resistance to BL/BLI in CRAB and *S. maltophilia* remain to be elucidated

CRAB and *S. maltophilia* are two nosocomial pathogens of increasing importance, particularly in the context of wide-spread use of broad-spectrum antimicrobials and increase in the size of the immunocompromised population at risk[6,101,102]. The treatment of infections caused by these organisms poses a significant challenge, as they harbor an extensive repertoire of chromosomal resistance determinants including multiple RND efflux pumps and beta-lactamases[103–107]. A handful of studies have analyzed the mechanisms of resistance to novel BL/BLI in these organisms, with particular attention to SUL/DUR[108] and FDR[100,109] in the case of CRAB, and FDR[27] in *S. maltophilia*. Efflux pump-based mechanisms of resistance were not identified in these studies; however, their involvement has not been excluded, and more work remains to done.

## Conclusions and future directions

The last decade has seen a much-needed increase in new antimicrobials for treating infections caused by resistant gram-negative organisms[7,110], in particular novel BL and BL/BLI agents. The wide-spread availability of genomic sequencing, in combination with in vitro evolution experiments, is also enabling a proactive approach in seeking potentially relevant resistance mechanisms that might play a role when these new agents enter clinical

practice. Emerging data reviewed here suggest that both major and lesser-known RND efflux pumps represent important resistance mechanisms in critical gram-negative bacterial pathogens. We believe that the roles of RND efflux pumps in resistance to novel BL/BLI agents merit more attention, and that broader genomic sequencing of clinical isolates will provide valuable insights into currently understudied, but important mechanisms.

In addition to genomic sequencing, work involving direct genetic manipulation is needed to establish the functional roles (or lack thereof) of the now large catalog of identified efflux pump mutations in clinical BL/BLI-resistant isolates. As reviewed extensively above, a primary weakness of many current studies is the observational nature of associations between mutations in efflux pump systems, often occurring in complex genetic backgrounds, and BL/BLI resistance. Construction of isogenic mutant series in which the direct consequences of individual changes can be observed is required to prove causality. In-frame chromosomal deletions of efflux pumps with direct MIC readout are an additional valuable approach to establish the contributions of individual pumps to resistance occurring in clinical isolates. These studies remain limited, as many labs that are specialized in genomic sequencing lack the capacity to perform genetic engineering, and additionally, genetic manipulation of wild type clinical isolates is often extremely challenging. However, establishment of a verified catalog of resistance mutations may facilitate direct molecular testing in the clinical diagnostic lab. Such testing may direct future targeted therapy with pump inhibitors.

Work is also required to determine the absolute transport specificities of RND efflux pumps contributing to resistance to BL/BLI combination agents. It has been assumed that efflux-mediated BL/BLI resistance is primarily due to transport of the beta lactam antibiotic component. But for efflux pumps that efficiently transport beta lactam antibiotics, it is very reasonable to assume that they may also transport structurally related beta lactam class beta lactamase inhibitors, such as tazobactam, but this is currently uncharacterized. It also unknown whether these same pumps are capable of transporting the structurally dissimilar diazabicyclooctane class beta lactamase inhibitors, such as avibactam, vaborbactam, and relebactam, though this seems less likely. Direct assays that test transport specificities or ligand binding assays are required but are technically challenging to perform. Another related challenge is the relative lack of direct structural data for many of the RND class transporters reviewed above. This is due in part to the general difficulty of structural studies on large, multiprotein complexes that span two membranes of different lipid composition.

Clear establishment of roles for specific alterations in specific transporters in mediating resistance will likely drive renewed structural and functional interest in these proteins. Structures and structural-functional studies will, in turn, stimulate the development of new antibiotics that are poor substrates for wild type and altered efflux pumps. They will also facilitate the development of targeted efflux pump inhibitors that may be co-administered in a manner analogous to beta lactamase inhibitors. We believe the many avenues of this work will yield both new antibiotic targets that are needed to maintain clinical efficacy of BL/BLI combination antibiotics, as well as new strategies to minimize the emergence of resistance.

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

## Acknowledgements
This work was supported by the Intramural Research Program of the National Institute of Infectious Diseases. The content and views expressed in this work are those of the authors and do not necessarily represent the official views of the NIH or the U.S. Government.

## Author contributions
A.D.C. and J.P.D. conceived of this work. A.D.C. and J.P.D. co-wrote the manuscript. ADC created Table 1 and critically assessed the associated literature cited within it. J.P.D. created Fig. 1. Both authors critically reviewed and edited the final version of the manuscript.

## Funding

## Competing interests
The authors declare no competing interests.
