## [Peer Review File · Communications Medicine]

Reviewers' comments:

Reviewer #1 (Remarks to the Author):

The manuscript "Recent insights into RND efflux pump-mediated resistance to novel beta lactam and beta lactam/beta lactamase inhibitor combination antibiotics in gammaproteobacteria" provides an overview of the literature on the contribution of efflux to resistance to the recent BL/BLI combinations. I think this is a nicely written manuscript and addresses an area of research into BL/BLI resistance that is largely understudied, and subsequently will be of interest to readers of this journal. I just have a few minor comments to make.

1. Line 29. The "(BL/BLI)" should be added here,
2. Line 61. "Pseudomonas" should be written out in full here.
3. Line 77. "Carbapenem" doesn't need to be capitalised.
4. Line 86. Although there are currently no EPs for clinical use I think it might be important to note that there are a number under development and in trials.
5. Line 112. I think it would be good to note here, instead of at line 152, that PDC is an ampC and intrinsic in *P. aeruginosa*.
6. Line 118. Can you define MMR?
7. Lines 149 and 200. I think you are referring to the "OprD" protein here.
8. Line 223. RamA and RamR aren't actually found in *E. coli*, but are present in other Enterobacteriales including *Klebsiella pneumoniae* and *Enterobacter cloacae*. It may be better to say "transcriptional factors found in Enterobacteriales" or something similar.
9. Lines 227 -229. I don't understand why here you list just one component of some of the efflux pumps, eg AcrF instead of AcrEF.
10. Lines 268-279. FDR resistance has also been linked to CZA-resistant KPC variants. <https://pubmed.ncbi.nlm.nih.gov/33915286/>

Reviewer #2 (Remarks to the Author):

This review presents a current picture of the state of understanding of the relationship of RND efflux pumps to antibiotic resistance, principally beta-lactams and BL/BLI combinations, in Gram-negative bacteria.

The writing style is clear and almost conversational (in a good way) and the information presented is a good overview of the understanding of efflux roles in resistance, which definitely vary with the specific antibiotic or combination. A number of the newer specific BLs and BL/BLI combinations are carefully reviewed. But, it is hard to make general conclusions about RND pumps based on the individual diverse cases. Is it at all possible to propose possible solutions for AMR based on efflux inhibition? The authors conclude that more study with newest tools is needed. This is clearly true – the field is quite understudied.

In the area of BL/BLI combinations, one factor that is insufficiently addressed might be focused on: is there enough study of the BLIs as efflux substrates? It is mentioned in connection with certain BL/BLI combinations, such as IMI/REL (line 173) where it is noted that REL efflux probably plays a role in resistance to IMI/REL. Assaying for BLI efflux is obviously harder than looking at MICs – but it could be addressed, at least in the conclusions. Wouldn't BLIs that are poor efflux substrates be useful in combinations?

Reviewer #3 (Remarks to the Author):

This manuscript is academically straightforward and strong. However, as a Perspective piece, it is I fear unlikely to exert a strong and necessary impact on the communities it is targeting, ie medical doctors and perhaps pharmaceutical company researchers, in its present form. Indeed, the authors present essentially a list of mutations within efflux pump components that have led major Gram negative human pathogens to resist novel clinically available BL and BL+BLI combinations. Their major scientific point is that, just like for conventional BLs, efflux pump play a major role in

resistance to novel BL and BL+BLI combinations, although specific mutations more often than not differ. While this is an important message, I think more can perhaps be made of the data they present. My suggestions are as follows:

-In the Table, indicate when every new drug product entered clinical use and when resistance to it was first detected (during clinical development, after reaching the clinic, during laboratory experiments or clinically post drug approval), what the estimated frequency of resistance is and what its clinical prevalence is. Indicate also if mutations were found in patients being treated with the drug product and/or upon an in vitro resistance evolution experiment. Highlight any mutations found in strains already resistant to conventional or other novel BLs or BL+BLI combinations (cite FoRs and prevalences if known).

-In the Figure, for those mutations identified in either environment and confirmed by isolation into an otherwise wild-type background, for each drug-product where sufficient data exist, identify and map their location on the canonical efflux pump complex. Mark any mutation pairs or combinations often found together or having seemingly evolved from one another.

-Modify the text to indicate any novel conclusions that emerge from this additional analysis with this specific question in mind: is there any evidence in vitro FoRs are higher and clinical resistance emergence faster for these novel BLs or BL + BLI combinations than they were initially for conventional BLs, including the firsts in class (penicillin, cephalosporin etc)? If so, can this be attributed to the increasingly large pool of pre-existing mutations, such as these mapped here to efflux pumps? And what, if anything, does that imply for these drug products and those that make & use them?

Reviewer #4 (Remarks to the Author):

In this article, Chaing and Dekker review the literature surrounding the influence of RND efflux pumps on resistance to beta-beta/lactamase inhibitors.

The article is well written. Table and figures are appropriate.

Major comments:

1. I think that the review would benefit from being more specific in which molecule efflux is believed to be acting on in the examples provided (e.g. for ceftazidime / avibactam, is it efflux of one or both molecules that is believed to be driving this observation in MICs/resistance?).
2. What are the potential therapeutic solutions to efflux if this is going to be an emerging problem with novel BL/BLI?

Response to Reviewers

Recent insights into RND efflux pump-mediated resistance to novel beta lactam and beta lactam/beta lactamase inhibitor combination antibiotics in gammaproteobacteria

Dulanto Chiang A and Dekker JP

Manuscript: COMMSMED-22-0369A

Reviewer #1:

The manuscript “Recent insights into RND efflux pump-mediated resistance to novel beta lactam and beta lactam/beta lactamase inhibitor combination antibiotics in gammaproteobacteria” provides an overview of the literature on the contribution of efflux to resistance to the recent BL/BLI combinations. I think this is a nicely written manuscript and addresses an area of research into BL/BLI resistance that is largely understudied, and subsequently will be of interest to readers of this journal. I just have a few minor comments to make.

Response: We thank this reviewer for their very careful review, positive assessment of the manuscript, and specific suggestions, which we have addressed below.

1. Line 29. The “(BL/BLI)” should be added here	Corrected.
2. Line 61. “Pseudomonas” should be written out in full here.	Corrected.
3. Line 77. “Carbapenem” doesn’t need to be capitalised.	Corrected.
4. Line 86. Although there are currently no EPs for clinical use I think it might be important to note that there are a number under development and in trials.	We have modified the sentence to: “Importantly, there are currently no standard approaches for specific molecular or functional testing for efflux-mediated resistance in clinical microbiology laboratories, and while efflux pump inhibitors are under development, none are currently available for clinical use.”
5. Line 112. I think it would be good to note here, instead of at line 152, that PDC is an ampC and intrinsic in P. aeruginosa .	We have modified this sentence to: Line 112: “The most common mechanisms of CZA and C/T resistance in P. aeruginosa are mutations in the chromosomal AmpC cephalosporinase PDC”

	Line 152: “Nonetheless, IMI can still be hydrolyzed to some extent by the P. aeruginosa PDC”
6. Line 118. Can you define MMR?	We have now spelled out MMR: Line 118: “These mutations spontaneously emerged in a DNA mismatch repair-deficient hypermutator background with a non-functional MexAB-OprM, and their role in conferring a 4- to 6-fold increase in minimum inhibitory concentration (MIC) to CZA, C/T and FEP was confirmed through genetic engineering in a wild type background.”
7. Lines 149 and 200. I think you are referring to the “OprD” protein here.	We have changed the name to OprD in both lines 149 and 200.
8. Line 223. RamA and RamR aren’t actually found in E. coli , but are present in other Enterobacterales including Klebsiella pneumoniae and Enterobacter cloacae . It may be better to say “transcriptional factors found in Enterobacterales” or something similar.	Thank you for catching this error. The sentence has been reworded: Line 223: “In many Enterobacterales, AcrAB-TolC is under the control of several regulators including the local repressor AcrR and global transcriptional factors MarR, MarA, RamR, RamA, SoxS and Rob.”
9. Lines 227 -229. I don’t understand why here you list just one component of some of the efflux pumps, eg AcrF instead of AcrEF.	We thank the reviewer for raising this point and realize how it was confusing. We have reworded and restricted to indicate the pumps with experimentally established compositions. Line 227-229: “Several additional RND efflux pumps are present in Enterobacterales, including AcrEF, and MdtABC in E. coli , and OqxAB, and KpgABC in K. pneumoniae , but the roles of these additional pumps in antibiotic resistance, as well as those of others outside the RND family, have not been studied in great depth”
10. Lines 268-279. FDR resistance has also been linked to CZA-resistant KPC variants. https://pubmed.ncbi.nlm.nih.gov/33915286/	Thank you for pointing out this oversight. We have added the reference to the manuscript. Line 268-279: “The FDR resistance mechanisms are still being characterized in Enterobacterales.

	Several in vitro and in vivo studies point to TonB-dependent iron transport mechanisms, including CirA and Fiu ^{90,91} , while others have reported substitutions/deletions in the AmpC beta-lactamase R2 loop ^{25,92} or the contribution of KPC alleles associated with CZA resistance ⁹³ .”
--	---

Reviewer #2:

This review presents a current picture of the state of understanding of the relationship of RND efflux pumps to antibiotic resistance, principally beta-lactams and BL/BLI combinations, in Gram-negative bacteria.

The writing style is clear and almost conversational (in a good way) and the information presented is a good overview of the understanding of efflux roles in resistance, which definitely vary with the specific antibiotic or combination. A number of the newer specific BLs and BL/BLI combinations are carefully reviewed. But, it is hard to make general conclusions about RND pumps based on the individual diverse cases. Is it at all possible to propose possible solutions for AMR based on efflux inhibition? The authors conclude that more study with newest tools is needed. This is clearly true – the field is quite understudied.

Response: We appreciate the reviewer’s very careful consideration of our manuscript, positive assessment, and specific and insightful comments. We believe the field is indeed understudied and it remains to be seen how well efflux pump inhibitors will work as antibiotics.

In the area of BL/BLI combinations, one factor that is insufficiently addressed might be focused on: is there enough study of the BLIs as efflux substrates? It is mentioned in connection with certain BL/BLI combinations, such as IMI/REL (line 173) where it is noted that REL efflux probably plays a role in resistance to IMI/REL. Assaying for BLI efflux is obviously harder than looking at MICs – but it could be addressed, at least in the conclusions. Wouldn’t BLIs that are poor efflux substrates be useful in combinations?

Response: We thank the reviewer for raising this crucial question. Indeed, there are few rigorous studies characterizing substrate specificity profiles for most of the major RND pumps, and this represents a major gap in current knowledge. If detailed specificities were known, this information could conceivably be used for combining existing BLIs and BL/BLIs into new combinations to thwart efflux mechanisms. Studies into efflux specificity could also inform the design of new drugs that are poor efflux substrates. We have added an additional recent reference on IMI/REL and efflux-mediated resistance to line 173 (PMID 36918743) and we have modified the conclusions (Line 304-).

Reviewer #3:

This manuscript is academically straightforward and strong. However, as a Perspective piece, it is I fear unlikely to exert a strong and necessary impact on the communities it is targeting, ie medical doctors and perhaps pharmaceutical company researchers, in its present form. Indeed, the authors present essentially a list of mutations within efflux pump components that have led major Gram-negative human pathogens to resist novel clinically available BL and BL+BLI combinations. Their major scientific point is that, just like for conventional BLs, efflux pump play a major role in resistance to novel BL and BL+BLI combinations, although specific mutations more often than not differ. While this is an important message, I think more can perhaps be made of the data they present.

Response: We thank this reviewer for their in-depth review of this manuscript and many valuable suggestions. Below we have addressed the comments to the best of our ability, given the substantial limitation that data to answer many of these important questions are unfortunately scarce.

In the Table, indicate when every new drug product entered clinical use	The dates of FDA approval (when applicable) have been added to the table.
and when resistance to it was first detected (during clinical development, after reaching the clinic, during laboratory experiments or clinically post drug approval), what the estimated frequency of resistance is and what its clinical prevalence is.	We agree with the reviewer that these are important questions. However, this is not trivial for a few reasons. First, we have access only to published records. Many or most cases of antibiotic resistance are not published. Second, in many of these published cases, the mechanism of resistance is not experimentally or rigorously established. Much resistance to the antibiotics we have considered in this review are NOT due to efflux pumps, and we do not want to misattribute overall trends in resistance that have not been characterized at the molecular level (due to mutations in beta lactamases, acquired carbapenemases, etc) to efflux pumps in this review without rigorous evidence. A significant driver of large scale resistance tends over time has been the global spread of specific bacterial clones, which often contain multiple resistance mutations and genes. The clonal spread of resistant strains is an important confounder to the question of increased efflux mediated

mechanisms when looking at global trends with isolates in which resistance mechanisms have not been rigorously characterized.

Regarding the time to actual first detection and report of resistance, resistant isolates are routinely identified for antibiotics during pre-clinical testing against panels of organisms. While statistics on resistant isolates are sometimes available in FDA submissions, often detailed characterization of genetic mechanisms of resistance are not.

For reporting mechanisms and resistance from the literature, there is a bias toward previously known resistance mechanisms, where in many genomic characterizations, only mutations in previously described major resistance genes are described. Because many RND efflux pumps are not fully characterized, we believe the literature substantially under-represents changes in many of these genes.

While the clinical prevalence of resistance to BL/BLI can be obtained from public MIC databases such as EUCAST (<https://mic.eucast.org/>), these numbers also represent all-comer resistance, and we do not have the data in most cases to be able to estimate what proportion of this resistance is due to efflux mechanisms in almost any case. In this course of this work, we surveyed the literature exhaustively, and there are very little actual quantitative data that would allow calculation of a percentage of resistant isolates, efflux plays a role.

Taking the above into consideration, we believe it is unfortunately not possible to indicate for any of the antibiotics we have included in our review an accurate assessment of when efflux-based resistance first emerged, though we agree entirely with the reviewer that this information would add a great deal if it were available.

Highlight any mutations found in strains already resistant to conventional or other novel BLs or BL+BLI combinations (cite FoRs and prevalences if known).	Mutations conferring resistance to novel BL/BLI in clinical isolates are usually described in the background of known multiresistant strains, as these are the organisms in which BL/BLI are used in clinical practice. Additionally, experimental validation of individual efflux pump mutations that confer resistance in wild-type strains has been very seldom performed, and thus, the conclusions that can be drawn in clinical isolates have to be taken with that context in mind. A second issue in the literature on clinical isolates is that lesser-known efflux pump mutations and known resistance variants are often identified simultaneously, often making it impossible to determine which emerged first, due to lack of information required for correct phylogenetic inference. We have included the most rigorous examples (in our view) in the text of our manuscript. Gomis-Font et al. (2022) identified a C/T resistant isolate carrying both MexCD-OprJ mutations and an OprD deletion. It is not known which emerged first, and the contribution of the OprD deletion was not validated experimentally. Similarly, Sadek et al (2023) described the progressive development of cefiderocol resistance in P. aeruginosa serial patient isolates from a CF patient. The genetic changes were described, which included mutations associated to the MexAB-OprM efflux pump, but the individual mutations did not undergo experimental validation on a wild-type background. Unfortunately in a review such as this we are limited to the work that has been published, and there simply isn't more that we are aware of that has not already been included in the manuscript.
In the Figure, for those mutations identified in either environment and confirmed by isolation into an otherwise wild-type	We thank the reviewer for raising this point and agree that this analysis will add value to the review. However, there are two important

background, for each drug-product where sufficient data exist, identify and map their location on the canonical efflux pump complex. Mark any mutation pairs or combinations often found together or having seemingly evolved from one another	points: (1) The majority of the mutations identified in the literature associated with BL/BLI resistance and cited in our review have not been confirmed by engineering into WT isolates, with the exception of the cited subset that have been validated in MexAB-OprM. (2) There are limited available high resolution structures for the relevant RND efflux pump assemblies. No high resolution assembled structures exist for MexXY-OprM, MexEF-OprN, MexCD-OprJ, MexGHI-OpmD, MexMN-OprM, MexPQ-OpmE, or MexVW-OprM, and thus we do not have a reference to which to map mutations or even determine structural homology for these other pumps, and this review is not intended for the deeper computational modeling that would be required to do this in a sufficiently rigorous manner. A high quality structure exists for MexAB-OprM, and we now have mapped the mutations with highest quality of evidence from Table 1 onto this 3D structure in a redesigned figure 1, and believe this adds significantly to the review.
Modify the text to indicate any novel conclusions that emerge from this additional analysis with this specific question in mind: is there any evidence in vitro FoRs are higher and clinical resistance emergence faster for these novel BLs or BL + BLI combinations than they were initially for conventional BLs, including the firsts in class (penicillin, cephalosporin etc)? If so, can this be attributed to the increasingly large pool of pre-existing mutations, such as these mapped here to efflux pumps? And what, if anything, does that imply for these drug products and those that make & use them	We thank the reviewer for this suggestion. While we'd love to answer the specific question of whether in vitro FoRs are higher or that clinical resistance has emerged faster for novel BLs and BL/BLIs than for older BLs, our assessment is that the data are simply too sparse and too messy to come to any conclusions here that we could state. Additionally, as we indicated above, the intention of the review is to focus on efflux mechanisms and much of the (messy and sparse) data on rates of resistance do not separate efflux and non-efflux-based mechanisms. Because of this lack of data, we don't really think we can comment in an evidence-based manner on whether the published pool of pre-existing efflux pump mutations that we have surveyed in this work are contributing to faster or higher rates of resistance to new BL and BL/BLI drugs vs. conventional BLs. We have added to the conclusion suggestions of future strategies

	involving the development of antibiotics that are poor substrates for the characterized efflux pumps and the use of targeted efflux pump inhibitors in combination with BL and BL/BLI.
--	--

Reviewer #4:

In this article, Chaing and Dekker review the literature surrounding the influence of RND efflux pumps on resistance to beta-beta/lactamase inhibitors.

The article is well written. Table and figures are appropriate.

Response: We thank the reviewer for their detailed review of our work, positive assessment and helpful comments, which are addressed below.

Major comments:

1. I think that the review would benefit from being more specific in which molecule efflux is believed to be acting on in the examples provided (e.g. for ceftazidime / avibactam, is it efflux of one or both molecules that is believed to be driving this observation in MICs/resistance?).	We thank the reviewer for this question, which is an important but complex one. The simple answer is that for many of the BL/BLI combinations, this has not been rigorously assessed experimentally, so there are not good data regarding which molecules – the BL and/or the BLI – are being transported by the efflux pump. This is also not entirely as straight forward as just assessing substrate specificity of these fairly nonselective pumps, as the contributions of efflux vs. other resistance mechanisms may depend on whether the BL is given with or without the BLI. For instance, in the case of ceftazidime administered by itself, a principal mechanism of resistance is usually hydrolysis by the cellular beta-lactamases such as AmpC, and to a much lesser degree efflux by RND class pumps such as MexAB-OprM. However, in presence of a BLI such as avibactam, the effect of the beta-lactamase is inhibited and efflux of ceftazidime plays a predominant role in conferring resistance, but it is not known whether avibactam is also effluxed by MexAB-OprM, and if so, to what extent this secondary efflux contributes to the measured MIC. These are all experimentally addressable in the right system, but
---	--

	unfortunately rigorous data that we can cite in this review do not exist for many of these molecules.
2. What are the potential therapeutic solutions to efflux if this is going to be an emerging problem with novel BL/BLI?	We thank the reviewer for this question. This is an area currently under study. Two potential approaches would be the simultaneous administration of efflux pump inhibitors with the antibiotic; or the development of new antibiotics that are poor substrates for major efflux pumps. We have modified the concluding sentence: Line 304: “This work may yield both new antibiotic targets as well as strategies to minimize the emergence of resistance, such as the development of new antibiotics that are poor substrates for characterized efflux pumps and co-administration with specifically targeted efflux pump inhibitors.”

Reviewers' comments:

Reviewer #1 (Remarks to the Author):

I think this is a very interesting manuscript and the current literature on this subject area has been very well reviewed/summarised. All of the previous reviewer comments have been addressed well and I have no further recommendations for modifications to this manuscript.

Reviewer #2 (Remarks to the Author):

The authors of this revised manuscript have addressed, either in the text or in the rebuttal letter, all of the points raised by the reviewers, including myself. In most cases, the questions raised by reviewers were appreciated by the authors but the extant data was insufficient to provide answers.

I think the whole review provides an excellent perspective on the state of research into the area of efflux effects on BL/BLI combinations – and it is clear that the area is under-studied and many questions are outstanding, really demanding further research. The authors have provided a clear understanding of the state of the art.

As noted in the reviews – this is a very well written review; it is eminently readable, easy to follow. I am impressed with the rigor the authors applied in selecting their references, being very selective in the quality of data presented.

It is notable that the authors have relied on genetic evidence where it is available – and not extended this to include implications from studies with efflux pump inhibitors – since in most cases, the totality of mechanisms of action of EPIs is unknown and such studies are generally complicated by non-specific mechanisms such as permeabilization.

I am very favorably impressed with this paper; it would be a strong contributor to the understanding of efflux effects on the many new BL/BLIs – for researchers, drug discoverers, and ID physicians.

Reviewer #3 (Remarks to the Author):

I want to thank the authors for having so carefully thought through my comments and for having generated figure a new Fig. 1 which I do believe does add substantially to the review, which should now in my view simply be accepted for publication.

I totally agree with what they say about the lack of published data presently to answer most of the questions I asked. However, I hope they agree that the new Fig. 1 gives a tantalising of glimpse of what might emerge in terms of new information if some of my questions were clinically investigated and/or experimentally addressed. I also want to add that I entirely agree that a lot of the data to answer those questions may already be available to researchers in industry. And if it isn't, it might best be obtained collaboratively with them. Indeed, I believe the Wellcome Trust-established SEDRIC is already trying to make such data available publicly (<https://wellcomeopenresearch.org/articles/3-59/v1>). So, if they aren't already, I encourage the authors to become SEDRIC members.

Response to Reviewers

RND efflux pump-mediated resistance to beta lactam and beta lactam/beta lactamase inhibitor combination antibiotics in gammaproteobacterial

Dulanto Chiang A and Dekker JP

Manuscript: COMMSMED-22-0369C

Reviewer #1:

I think this is a very interesting manuscript and the current literature on this subject area has been very well reviewed/summarised. All of the previous reviewer comments have been addressed well and I have no further recommendations for modifications to this manuscript.

Response: We thank this reviewer sincerely for their initial very careful review, and positive assessment of the revised manuscript.

Reviewer #2:

The authors of this revised manuscript have addressed, either in the text or in the rebuttal letter, all of the points raised by the reviewers, including myself. In most cases, the questions raised by reviewers were appreciated by the authors but the extant data was insufficient to provide answers.

I think the whole review provides an excellent perspective on the state of research into the area of efflux effects on BL/BLI combinations – and it is clear that the area is under-studied and many questions are outstanding, really demanding further research. The authors have provided a clear understanding of the state of the art.

As noted in the reviews – this is a very well written review; it is eminently readable, easy to follow. I am impressed with the rigor the authors applied in selecting their references, being very selective in the quality of data presented.

It is notable that the authors have relied on genetic evidence where it is available – and not extended this to include implications from studies with efflux pump inhibitors – since in most cases, the totality of mechanisms of action of EPs is unknown and such studies are generally complicated by non-specific mechanisms such as permeabilization.

I am very favorably impressed with this paper; it would be a strong contributor to the understanding of efflux effects on the many new BL/BLIs – for researchers, drug discoverers, and ID physicians.

Response: We thank this reviewer for the very careful review, extremely helpful comments, and positive assessment of the revised manuscript.

Reviewer #3:

I want to thank the authors for having so carefully thought through my comments and for having generated figure a new Fig. 1 which I do believe does add substantially to the review, which should now in my view simply be accepted for publication.

I totally agree with what they say about the lack of published data presently to answer most of the questions I asked. However, I hope they agree that the new Fig. 1 gives a tantalising of glimpse of what might emerge in terms of new information if some of my questions were clinically investigated and/or experimentally addressed. I also want to add that I entirely agree that a lot of the data to answer those questions may already be available to researchers in industry. And if it isn't, it might best be obtained collaboratively with them. Indeed, I believe the Wellcome Trust-established SEDRIC is already trying to make such data available publicly (<https://wellcomeopenresearch.org/articles/3-59/v1>). So, if they aren't already, I encourage the authors to become SEDRIC members.

Reviewer #4 (original comments, further answered per request from Editor):

In this article, Chaing and Dekker review the literature surrounding the influence of RND efflux pumps on resistance to beta-beta/lactamase inhibitors.

The article is well written. Table and figures are appropriate.

Major comments:

1. I think that the review would benefit from being more specific in which molecule efflux is believed to be acting on in the examples provided (e.g. for ceftazidime / avibactam, is it efflux of one or both molecules that is believed to be driving this observation in MICs/resistance?).

Response: This is now discussed in lines 448-454 (tracked change version).

2. What are the potential therapeutic solutions to efflux if this is going to be an emerging problem with novel BL/BLI?

Response: This is now discussed in lines 461-463 (tracked change version).